# First Reported Circulation of Equine Influenza H3N8 Florida Clade 1 Virus in Horses in Italy

**DOI:** 10.3390/ani14040598

**Published:** 2024-02-12

**Authors:** Ida Ricci, Silvia Tofani, Davide Lelli, Giacomo Vincifori, Francesca Rosone, Andrea Carvelli, Elena Lavinia Diaconu, Davide La Rocca, Giuseppe Manna, Samanta Sabatini, Donatella Costantini, Raffaella Conti, Giulia Pacchiarotti, Maria Teresa Scicluna

**Affiliations:** 1Istituto Zooprofilattico Sperimentale del Lazio e della Toscana “M. Aleandri”, Via Appia Nuova 1411, 00178 Rome, Italy; ida.ricci@izslt.it (I.R.); francesca.rosone@izslt.it (F.R.); andrea.carvelli@izslt.it (A.C.); elena.diaconu@izslt.it (E.L.D.); davide.larocca@izslt.it (D.L.R.); giuseppe.manna@izslt.it (G.M.); samanta.sabatini@izslt.it (S.S.); donatella.costantini@izslt.it (D.C.); raffaella.conti@izslt.it (R.C.); giulia.pacchiarotti-esterno@izslt.it (G.P.); teresa.scicluna@izslt.it (M.T.S.); 2Istituto Zooprofilattico Sperimentale della Lombardia e dell’Emilia Romagna “Bruno Ubertini”, Via Bianchi, 9, 25124 Brescia, Italy; davide.lelli@izsler.it; 3Istituto Zooprofilattico Sperimentale dell’Abruzzo e del Molise “G. Caporale”, Campo Boario, 64100 Teramo, Italy; g.vincifori@izs.it

**Keywords:** equine influenza, Florida clade 1, horse, genetic characterization HA and NA

## Abstract

**Simple Summary:**

Equine influenza (EI) is an acute and highly contagious viral disease of equids characterized by fever and respiratory signs. Ongoing antigenic mutations that are typical of influenza viruses, which may cause a reduction in the effectiveness of vaccines, highlight how crucial both surveillance and virus characterization are for updating vaccine compositions. The aim of this study was to verify the identity of the equine influenza virus strains detected in Italy, especially in the absence of formal active surveillance. Twenty nasal swabs, collected from symptomatic horses located in North and Central Italy between February and April 2019 were positive for influenza A virus (IAV) RRT-PCR. Sequencing identified an isolated strain as H3N8 Florida lineage clade 1 for one sample from Brescia, Lombardy Region. This study is the first report of H3N8 Florida lineage clade 1 circulation in Italy, confirming the value of monitoring for EIV circulating strain in relation to the appropriateness of the vaccine virus composition for maximum efficacy.

**Abstract:**

Background: Equine influenza (EI) is a highly contagious viral disease of equids characterized by pyrexia and respiratory signs. Like other influenza A viruses, antigenic drift or shift could lead to a vaccine-induced immunity breakdown if vaccine strains are not updated. The aim of this study was to genetically characterize EIV strains circulating in Italy, detected in PCR-positive samples collected from suspected cases, especially in the absence of formal active surveillance. Methods: Between February and April 2019, blood samples and nasal swabs collected from each of the 20 symptomatic horses from North and Central Italy were submitted to the National Reference Centre for Equine Diseases in Italy to confirm preliminary analysis performed by other laboratories. Results: None of the sera analysed using haemagglutination inhibition and single radial haemolysis presented a predominant serological reactivity pattern for any antigen employed. All nasal swabs were positive with IAV RRT-PCR. Only one strain, isolated in an embryonated chicken egg from a sample collected from a horse of a stable located in Brescia, Lombardy, was identified as H3N8 Florida lineage clade 1 (FC1). In the constructed phylogenetic trees, this strain is located within the FC1, together with the virus isolated in France in 2018 (MK501761). Conclusions: This study reports the first detection of H3N8 FC1 in Italy, highlighting the importance of monitoring circulating EIV strains to verify the vaccine composition appropriateness for maximum efficacy.

## 1. Introduction

Equine influenza (EI) is an acute, highly contagious viral disease of equids characterized by fever and respiratory signs especially dry coughing and serous nasal discharge, which is rarely fatal. Bacterial respiratory complications that could lead to a lethal outcome are however possible, especially under poor hygienic conditions. Equine influenza virus (EIV) is enveloped and has a segmented, negative-sense single-strand RNA genome. EIV is related to but distinct from the viruses that cause human and avian influenza [1]. EI is caused by two subtypes of the influenza A virus (family *Orthomyxoviridae*): H7N7 and H3N8. While the H7N7 has not been reported since 1980 [2], H3N8 subtypes still have major sanitary and economic impacts on the equine industry in various parts of the world [3], especially during gatherings of young animals of different origins together with adults with irregular vaccination and the use of non–updated vaccines. After circulating as a single lineage for at least two decades, during the mid-1980s, H3N8 split into two different lineages: American and European. Subsequently, the American lineage subdivided into three different sublineages—respectively named Kentucky, South American, and Florida—the last of which, since the 2000s, has become the predominant sublineage, diverging into clades 1 and 2 [4,5,6]. Florida clade 1 (FC1) strain was reported to circulate in North America and Florida clade 2 (FC2) in Asia and Europe. While FC2 was isolated once since 2019, in Europe, FC1 has been the predominant clade since 2019 [7,8,9,10]. In Italy, EIV circulation was reported several times between 1991 and 2014. Isolates of 1991 and 1999 clustered within the European lineage [11,12]. Isolates from outbreaks occurring during 2003 and 2004 in Rome and during 2005 in Bari were identified as members of the American lineage, belonging to H3N8 Florida clade 2, as those isolated in Rome in 2014 [8,11,13,14]. Like other influenza A viruses, EIV undergoes mutations within the HA antigen due to the antigenic shift or drift that may cause a reduction in the effectiveness of vaccines. Ongoing mutations of influenza viruses highlight how crucial both surveillance and virus characterization are for updating vaccine composition [14] as few but relevant amino acid changes in the HA antigen can lead to a significant diversity in antigenicity in the EIV. These changes can happen suddenly, highlighting the need to continuously monitor circulating field viruses [15]. In view of this, the World Organisation for Animal Health (WOAH) Expert Surveillance Panel (ESP) conducts a strain selection process based on genetic, antigenic, and epidemiological data collected and annually reported to the ESP to provide recommendations on which EI strains should be included in the vaccines [16]. Although in Italy there is no ongoing active surveillance, local field veterinarians refer flu-like signs where EIV circulation is suspected when submitting samples for laboratory analysis. In this study, we describe the virological and serological tests performed on nasal swabs and blood samples collected during 2019 in North and Central Italy (Lombardy and Molise regions) from horses with flu-like signs and the identification of the strain detected. In Italy, the horse industry is still an economically important sector especially due to substantial ongoing public funding.

## 2. Materials and Methods

### 2.1. Sample Collection

Between February and April 2019, nasal swabs and blood were collected from 20 horses presenting clinical signs of fever, anorexia, coughing, conjunctival and nasal discharge in stables, respectively located in North and Central Italy (Lombardy and Molise regions). The nasal swabs were placed in sterile saline or phosphate buffer solutions and after being analysed by the local laboratories with RT-PCR [17,18,19], these were forwarded to the National Reference Centre for Equine Diseases (CeRME), located at the Istituto Zooprofilattico Sperimentale Lazio e Toscana (IZSLT) for confirmation and characterization of the detected EIV.

### 2.2. Virological Tests

Viral RNA extraction was performed from nasal swabs using the QIAsymphony DSP Virus/Pathogen Kits by the automated extractor QIAsymphony SP (Qiagen, Valencia, CA, USA) according to the manufacturer’s instructions. The eluate was used for reverse-transcriptase real-time (RRT) PCR for the detection of the influenza A virus (IAV) [17] using the QuantStudio 7 Flex Real-Time PCR System (Applied Biosystem, Waltham, MA, USA). Samples with Ct < 40 were defined as EIV positive.

PCR-positive nasal swabs were inoculated (0.1 mL) into the allantoic cavities of 10-day embryonated specific-pathogen-free chicken eggs. After 48–72 h of incubation at 37 °C, the allantoic fluid was harvested and checked for the presence of a haemagglutinating virus employing the standard methods available [20]. A maximum of three successive passages were made and allantoic fluids with an HA titre equal to or greater than 1:2 were verified in EIV PCR [17], and when positive, used for virus genetic characterization.

### 2.3. Genetic Analysis

RNA from positive PCR samples was further characterized using Sanger sequencing of the haemagglutinin (HA) and neuraminidase (NA) genes using the amplification and sequencing primers as described elsewhere [21].

Both sense and anti-sense strands of the HA and NA PCR products were sequenced using Applied Biosystems BigDye version 3.1. (Life Technologies, Danvers, MA, USA) with the 3500 Genetic Analyzer sequencer (Applied Biosystems by Thermo Fisher Scientific, Foster City, CA, USA) and were acquired using the Sequencing Analysis software 7. The sequences were assembled and analysed using the Geneious Prime 2021.0.3 and then compared to other EIV HA and NA sequences using the BLAST web-based program (http://www.ncbi.nlm.nih.gov/BLAST, accessed on 1 September 2023).

### 2.4. Phylogenetic Analysis

Sequences obtained were aligned against the nucleotide (nt) database using blastn software both for HA and NA genes (see Appendix A). Multiple alignments were conducted using MUSCLE software (https://www.ebi.ac.uk/Tools/msa/muscle/, accessed on 1 September 2023) with standard parameters. HA and NA gene trees were built with Bayesian Markov chain Monte Carlo (MCMC) using MrBayes version 3.7.1 [22]. Instead of selecting a specific substitution model obtained from a model testing approach, sampling across the substitution model space in the Bayesian MCMC analysis [23] was applied. The topology, the branch lengths, the four stationary frequencies of the nucleotides, the six different nucleotide substitution rates, the proportion of invariable sites, and the shape parameter of the gamma distribution of rate variation were set as default in the prior model. Both phylogenetic trees were plotted in RStudio (4.3.0) [24] using ggtree and tidyverse libraries [25,26]. The amino acid substitutions were evaluated performing blastp, using as query the South Africa strain (USI49340.1 for HA and USI49342.1 for NA genes) and as subject a group of protein sequences composed of the Miami strain (ABY81492.1 for HA and ABY81495.1 for NA) and most recent circulating FC1 strains, including the one detected in this study.

### 2.5. Serological Tests

EIV antibody titres were determined for all 20 serum samples using the HI assay, according to the WOAH standard procedure [15]. Chicken erythrocytes at 1% were employed together with the reference antigens: A/eq/Praga/1/56 (H7N7), A/eq/Newmarket/2/93 (reference strains for the H3N8 European lineage), and A/eq/South Africa/04/2003 (reference strain for H3N8 American lineages) belonging to FC1, considering the virological results obtained [27]. A titre equal to or higher than the final dilution of 1:8 of the serum analysed is considered positive for the presence of haemoagglutinating antibodies against the antigen kindly supplied by WOAH NRL.

Serological testing was also conducted using single radial haemolysis (SRH), considered the gold standard for defining the level of protection to infection in relation to the area of haemolysis (mm^2^) measured [27]. The protocol employed was in accordance with that described in the WOAH Terrestrial Manual [20]. The standard control reference sera were provided by the European Directorate for the Quality of Medicines and Healthcare and were included in serological assays to standardize the results: A/eq/Praga/1/56 and A/eq/Newmarket/77 (H7N7), A/eq/Newmarket/2/93 (H3N8 European lineage) and A/eq/South Africa/04/2003 (H3N8 Florida sublineage clade 1). The reference antigens were the same as mentioned above for the HI. The criteria for the classification of the protection level related to the haemolysis area were categorized as negative (haemolysis area = 0 mm^2^), clinically not protected (haemolysis area < 85 mm^2^), clinically protected (haemolysis area 85 ≤ mm^2^ ≤ 150) and protected against infection (haemolysis area > 150 mm^2^) [27,28]. Fisher’s exact test (R software version 4.1.1) was performed to assess the presence of an association between the virus lineages and the SRH serological titres, classified into 4 categories based on the protection level [27,28]. Fisher’s exact test was performed among the three virus lineages (H7N7, H3N8 FC1, H3N8 Eu) and also between the two H3N8 virus lineages (H3N8 FC1, H3N8 Eu).

## 3. Results

### Virological Test

Of the 20 horses with respiratory signs, from which blood samples and nasal swabs were collected, 14 were located in three different stables in Campobasso, province of the Molise region and 6 in two stables, respectively located in the Brescia (5) and Bergamo (1) provinces of the Lombardy region. All the farms belonged to the equestrian sector and were classified as riding centres or had saddle horses that were stabled at the owner’s facilities. The morbidity rate, intended as the number of horses with clinical signs referable to IE, is reported in Table 1. The age and breed of positive horses are reported in Table 2. All horses were categorized as saddle horses. In the case of the outbreaks in North Italy, only one recent introduction of four horses from Cordoba was reported, four days prior to the onset of clinical signs in three of the horses. In Central Italy outbreaks, horses from the three positive farms had attended a traditional gathering of horses (Festa di San Biagio) a couple of weeks before the onset of the clinical signs. Hundreds of horses from the entire region gathered at this traditional event, which served as the most likely source of the outbreak.

Table 3 reports the results of the serological and virological tests performed.

For the serological results, three horses in the same stable were H7N7 positive in both HI and SRH tests.

Fifteen horses out of twenty (75%) in HI and 18/20 (90%) in SRH were H3N8 American FC 1 positive.

Twelve horses out of twenty (60%) in HI and 14/20 (70%) in SRH were positive for the H3N8 European lineage.

Thirteen horses out of sixteen that resulted positive for both H3N8 lineages had a wider area in the SRH test (Table 4).

A significant difference was observed between the three strains and the four protection levels (*p* < 0.001). No difference was observed comparing the two H3N8 sublineages (*p* = 0.4).

The nasal swabs of all the 20 subjects examined, coming from the five different stables in Lombardy (2) and Molise regions (3), were EIV positive using RRT-PCR [17] (Table 3). Only two among the twenty samples obtained from the harvesting of the allantoic fluid from the eggs inoculated with the nasal swabs of symptomatic subjects were positive in RRT-PCR. These positive samples were from two horses of the stable situated in Brescia, Lombardy region, serologically negative at the time of sampling and were possibly collected during the early stages of infection, considering the short-lived shedding of EIV.

Sequencing was successful for only one of these samples, Brescia/19038317_3/2019, and genetic analysis identified the EIV strain at the origin of the Brescia episode as an FC1 strain (Figure 1 and Figure 2).

Partial sequences of Brescia/19038317_3/2019 were submitted on Genbank under accession numbers OP926929 and OP926930 for HA and NA gene fragments, respectively.

The phylogenetic analysis involved 43 and 44 sequences for, respectively, HA (929 bp length) and NA (986 bp length), including those of Brescia/19038317_3/2019.

Both trees represented in Figure 1 and Figure 2 are drawn to scale, with branch lengths measured in the number of substitutions per site.

These consensus trees scored an average standard deviation of split frequencies of 0.013432 and of 0.015489 for HA and NA, respectively. Each consensus tree was built from 2404 trees in four files (sampling 1804 of them): each file contained 601 trees ((ngen/sample freq) + 1), of which 451 were sampled.

The genetic analysis defined that the EIV strain at the origin of the Brescia outbreak belongs to FC1 (Figure 1 and Figure 2). Table 5 reports the identity percentage values of the sequences from the present study for HA and NA sequences with their closest strains obtained from BLAST.

In the HA gene tree (Figure 1), the FC1 is made up of two major clusters, one of which includes the South Africa (ON797670.1) and Ohio (DQ124192.1) viruses, representing the earliest strains; while the other cluster includes the most recently detected strains, together with Brescia/19038317_3/2019 that is located near to Santiago (MH346720.1) and other European strains.

Other relevant clades are FC2, as described by the Newmarket (FJ375213.1) and the Richmond (FJ195395.3) sequences, Kentucky clade (L39914.1), Eurasian clade represented by Berlin (CY032413.1) and Grosbois (KY241313.1) sequences, and Pre-Divergent clade, identified by Miami (CY028836.1) California (CY028812.1) and France (KY241305.1) sequences (see Appendix A for the estimated marginal likelihoods for each run for the HA gene fragment).

In the NA consensus tree (Figure 2), similarly to the HA gene tree, the South Africa (ON797672.1) and Ohio (DQ124168.1) strains are representative of FC1, which is made up of two major clusters; again, the Brescia/19038317_3/2019 groups with the one from Chile (MH346583.1) and the other recently detected European strains.

Other clades are FC2, as described by the Richmond (KF559336.1) sequence, Eurasian clade as described by Berlin (CY032415.1) and Grosbois (KY241345.1) sequences, and Pre-Divergent clade as described by Miami (CY028838.1), California (CY028814.1), and France (KY241337.1) sequences. (see Appendix A for the estimated marginal likelihoods for each run for the NA gene fragment).

The number of amino acid substitutions identified between the South Africa strain, used as reference, and the group of protein sequences included in this comparison were few. Of note, as shown in Table 6 and Table 7, HA and NA protein sequences of Brescia/19038317_3/2019 exhibit a total of seven and nine amino acid substitutions, respectively. In terms of amino acid substitutions, most recent European strains differ from Brescia/19038317_3/2019 in four positions: 285, 289, 302, and 385. The second, the third, and the fourth are substitutions in which the amino acid polarity of those positions changes. For Brescia/19038317_3/2019 NA protein, its amino acid substitution trend is much more similar to those of most recent European strains except in I159V, which does not change the polarity in the specific position that is also reported in the Miami strain.

## 4. Discussion

Strain characterization using the serological positivity pattern obtained with the serum samples available was not possible owing to the lack of a predominant reaction against any of the specific antigens used in the test, denoting that the blood samples were probably collected at a too-early stage of infection for this purpose. This was further hindered by the impossibility of obtaining a second set of samples to evaluate a seroconversion. The absence of a specific reaction could have been masked either by previous vaccinations or natural infections. Indeed, vaccination history was unavailable for the horses included in the study, even if three horses from the Molise region had antibodies for H7N7, detected with HI and SRH, usually indicative of the use of vaccines that still contain this subtype, considering it has not been detected since 1980. In addition, there was no history of remote or recent flu infections in these stables.

Presumably, the episodes were still active due to the total positivity of the IAV RRT-PCR for all the samples examined, even if the Ct values were high, which could have even been due to the storage conditions of the samples and the stability of the viral nucleic acid of the EIV. Further evidence of an ongoing infection was also the success of isolation of EIV in swabs inoculated in the chicken eggs, even if limited to two samples in the still seronegative horses.

The EIV genetic characterization was for only one of the two samples for which viral isolation was still successful, viz., Brescia/19038317_3/2019, which when compared to other HA and NA EIV sequences using BLAST, confirmed the first report of the circulation of FC1 in Italy (Accession Number OP926929 and OP926930) [8]. This represented further evidence of the persistence of circulation of FC1 in Europe, as reported respectively in 2018 in France and 2019 in the United Kingdom (UK) [8,16,31].

The phylogenetic analysis confirms the results of the BLAST alignments for both trees, referable to the HA and NA genes. For FC1, for each tree, the sequences separate into two groups, one containing older sequences and the other with the more recently detected sequences from France, the UK, and Chile [8,9]. As shown in Figure 1 and Figure 2, although Brescia/19038317_3/2019 clusters together with the most recent European and South American sequences (MH346720.1 for HA and MH346583.1 for NA genes), in both cases they do not share the same node, suggesting the potential precursor role of the latter viruses in the subsequent European outbreaks.

Amino acid substitutions reported in Appendix A indicate a low substitution rate. Interestingly, the HA protein of Brescia/19038317_3/2019 retains a more similar amino acid profile to that of the Miami strain in three positions, two of which are inside (I289T) and in proximity (V285A) to the site C linear epitope of human influenza virus. This is important in consideration of an immune response as some amino acid substitutions can lead to a link with glycoproteins that could mask the site and increase the viral ability to escape the immune response [28]. For the amino acid substitutions detected for the NA, no information is available regarding their biological significance.

The predominance of FC1 in Europe since 2018 could be due to frequent international movement of horses for competition without proper quarantine measures, together with the immune status of the exposed horse population to the emerging virus [9]. In light of this evidence, vaccines used in Europe should contain both clade 1 and clade 2 viruses from the Florida sublineage with the most recent ones available from WOAH reference laboratories [32].

The surveillance of EI is essential to both manage outbreaks and limit viral spread and to obtain data on the antigenic drift and shift of the circulating strains for updating vaccines.

Given the tendency of these viruses to mutate, it is highly important that field veterinarians submit samples from suspected clinical cases to monitor and properly manage EI in the equine population, so as to obtain reliable virological and epidemiological information [9]. After humans, the horse is subjected to the most frequent movement amongst domestic animals, which underlines the importance of surveillance of the most relevant equine infectious diseases for their adequate management to limit their spread.

A formal worldwide equine influenza monitoring program has been in force since 1995, with the ESP collecting epidemiological information on EIV outbreaks, performing antigenic characterization of the circulating EIV isolates, and reporting the results to the WOAH for their dissemination and sharing with the scientific community [33].

As already stated, in Italy, there is currently no active formal surveillance system and there is no data on how widespread the infection may be [14]. Up to the promulgation of the Animal Health Law (AHL) [34], the former legislation in Italy [35] required the mandatory notification of equine influenza with consequent application of sanitary restrictions in case of outbreaks. As a result, there has been a reluctance to submit samples in cases where suspicion of infection occurs, leading to a likely underestimation of disease occurrence; thus, the cases reported in our study likely represent only a few cases amongst many, with a more widespread level of viral circulation likely present. In the AHL, equine influenza is not considered in the list of diseases for which provisions are laid down; however, there are equestrian events that require the mandatory vaccination of equine participants and that they do not present signs referable to EI. Furthermore, even if the AHL does not specifically refer to this infection, it can still be considered in the context of animal wellbeing.

## 5. Conclusions

At the end of 2018, the ESP confirmed an extensive epidemic in Europe that continued throughout 2019 in France, the UK, and Eire [16,31] involving both vaccinated and unvaccinated horses. Of note is that vaccinated horses showed mild to no clinical signs, probably due to the presence of cross-protective immunity, especially amongst those horses that had been regularly vaccinated over several years. Circulation of the virus and onset of outbreaks could be explained by the decrease in vaccination interventions that should have otherwise ensured protection from infection and disease. This confirms the importance of vaccination, which confers a certain level of protection that reduces clinical signs and the duration of viral shedding [9].

In conclusion, vaccination, together with biosecurity measures, is the cornerstone for the control and prevention of EIV outbreaks [9]. The genetic evolution of the virus may determine, in the end, an increase in the risk of vaccine inefficacy, with the need to review the vaccine antigens to maintain their immune-competence relative to the circulating strains.

In view of this, the WOAH Reference Laboratories and other laboratories around the world that collect and share data on other equine disease outbreaks are useful in raising awareness of the potential of introducing viruses of possible local introduction, together with strain characterization that is also fundamental for vaccine updating. Collaboration of data from these laboratories with the stakeholders in the equine industry is crucial to control and prevent equine infectious diseases including EI, considering the frequent intercontinental movements of horses for auctions, competition events, and breeding activities [9,33].

## Figures and Tables

**Figure 1 animals-14-00598-f001:**
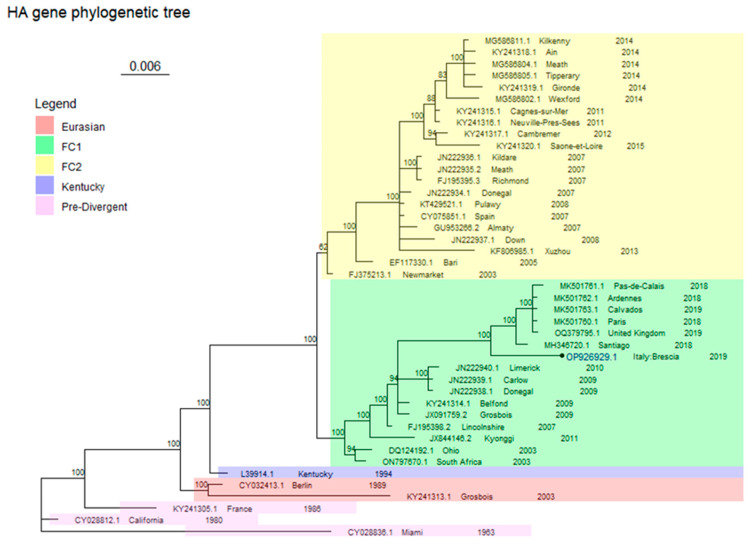
HA tree. Antigenic clades are coloured as shown in the legend. The probabilities of each clade node are expressed as percentages. Brescia/19038317_3/2019 has a round tip and its accession number is written in bold blue.

**Figure 2 animals-14-00598-f002:**
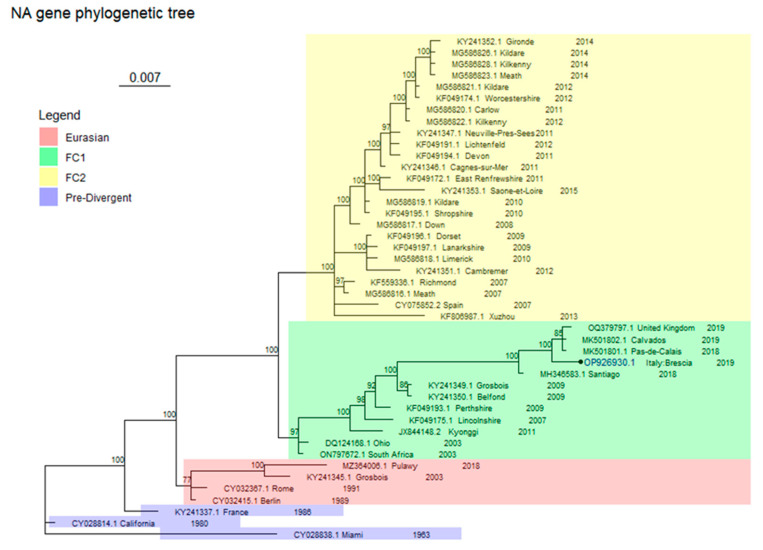
NA tree. Antigenic clades are coloured as shown in the legend. The probabilities of each clade node are expressed as percentages. Brescia/19038317_3/2019 has a round tip and its accession number is written in bold blue.

**Table 1 animals-14-00598-t001:** Morbidity rate (ND: not determined).

Farm ID	No. Horses with cl. Signs	No. of Positive Horses	No. of Stabled Horses	Morbidity Rate (%)
1	12	12	21	57.14%
2	1	1	2	50.00%
3	1	1	8	12.50%
4	7	5	52	9.62%
5	1	1	ND	ND

**Table 2 animals-14-00598-t002:** Age and breed of positive horses (TB: thoroughbred horse; AA: Anglo-Arabian horse; H: Haflinger horse; ND: not determined).

Region	Province	Farm ID	Horse ID	Age (Years)	Breed	Sex
Molise	Campobasso	1	1	8	TB	M
			2	7	AA	M
			3	4	TB	M
			4	10	TB	M
			5	9	TB	M
			6	11	AA	M
			7	6	ND	F
			8	ND	ND	ND
			9	5	TB	M
			10	8	TB	M
			11	11	TB	M
			12	ND	ND	ND
		2	13	9	ND	M
		3	14	4	TB	M
Lombardia	Brescia	4	15	9	H	M
			16	7	ND	F
			17	3	ND	M
			18	8	ND	M
			19	10	ND	ND
	Bergamo	5	20	8	ND	M

**Table 3 animals-14-00598-t003:** Serological and virological results obtained (* Titres reported in HI as reciprocal of the dilution of serum; ** Haemolysis diameter area in mm^2^; p°: level of protection, ++: protected against infection, +: clinically protected, - : clinically not protected, 0: negative; ^$^ titre obtained in HA).

Origin of Samples				Serological Results									Virological Results			
				Haemagglutination Inhibition *			Single Radial Haemolysis **						Nasal Swab		Allantoic Fluid	
Region	Province	Farm ID	Horse ID	H7N7	H3N8 Am FC1	H3N8 Eu	H7N7		H3N8 Am FC1		H3N8 Eu		HA	RRT PCR	HA	RRT PCR
							Area	p°	Area	p°	Area	p°				
Molise	Campobasso	1	1	<8	128	128	0	0	141	+	124	+	-	+	-	
			2	<8	64	128	0	0	149	+	164	++	-	+	-	
			3	<8	512	1024	0	0	304	++	272	++	-	+	-	
			4	128	128	64	121	+	233	++	186	++	-	+	-	
			5	<8	32	<8	0	0	170	++	40	-	-	+	-	
			6	<8	16	<8	0	0	72	-	0	0	-	+	-	
			7	<8	128	8	0	0	115	+	0	0	-	+	-	
			8	64	8	8	126	+	130	+	123	+	-	+	-	
			9	<8	256	256	0	0	263	++	263	++	-	+	-	
			10	128	16	64	127	+	163	++	164	++	-	+	-	
			11	<8	64	32	0	0	204	++	184	++	-	+	-	
			12	<8	64	<8	0	0	99	+	0	0	-	+	-	
		2	13	<8	16	<8	0	0	84	-	0	0	-	+	-	
		3	14	<8	128	128	0	0	223	++	179	++	-	+	-	
Lombardia	Brescia	4	15	<8	<8	<8	0	0	0	0	0	0	-	+	+ (32) ^$^	+
			16	<8	32	16	0	0	140	+	114	+	-	+	-	
			17	<8	<8	<8	0	0	0	0	0	0	-	+	+ (32) ^$^	+
			18	<8	<8	<8	0	0	94	+	118	+	-	+	-	
			19	<8	<8	8	0	0	82	-	119	+	-	+	-	
	Bergamo	5	20	<8	<8	<8	0	0	90	+	85	+	-	+	-	

**Table 4 animals-14-00598-t004:** Clinical protection categorization in relation to the SRH area diameter; protected against infection >150 mm^2^; clinically protected 85–150 mm^2^; clinically not protected < 85 mm^2^; negative 0 mm^2^ [27,28].

		Lineage		
Protection Level	SRH Area (mm^2^)	H7N7	H3N8 Am FC1	H3N8 Eu
Negative	0	17	2	6
Not protected	0 < mm^2^ < 85	0	3	1
Clinically protected	85 ≤ mm^2^ ≤ 150	3	8	6
Protected against infection	>150	0	7	7

**Table 5 animals-14-00598-t005:** BLAST results obtained by comparing the query sequences of Brescia/19038317_3/2019 (for HA and NA fragments) to the most similar strains filtered by accession number (A.N.), query coverage (Query cov.), E value, and identity percentage.

Brescia/19038317_3/2019 HA (OP926929.1)					Brescia/19038317_3/2019 NA (OP926930.1)			
Location	A.N.	Query Cov.	E Value	% Identity	A.N.	Query Cov.	E Value	% Identity
Santiago	MH346720.1	100%	0	98.49	MH346583.1	100%	0	98.99
Paris	MK501760.1	100%	0	98.39	/	/	/	/
U.K.	OQ379795.1	100%	0	98.39	OQ379797.1	100%	0	99.29
Pas-de-Calais	MK501761.1	100%	0	98.28	MK501801.1	100%	0	99.39
Ardennes	MK501762.1	100%	0	98.28	/	/	/	/
Calvados	MK501763.1	100%	0	98.28	MK501763.1	100%	0	99.39

**Table 6 animals-14-00598-t006:** HA differences in amino acid terms between the South Africa strain and a group of protein sequences composed of the Miami strain and most recent circulating FC1 strains. Asterisk indicates a.a located within a linear epitope identified in a human flu virus [29,30].

Amino Acid Position		236	255	265	273	274	280	285	289 *	302	322	323	325	337	385	400	463	467	477	487	492	502	516
USI49340.1	South africa	V	I	V	L	K	V	V	I	S	I	R	N	P	A	R	K	Q	G	D	G	Y	K
ABA39846.1	Ohio	.	.	.	.	.	.	.	.	.	.	.	.	.	.	.	.	.	.	.	.	.	.
WCL41102.1	UK	I	.	.	.	.	.	.	.	.	.	.	.	.	T	.	R	L	.	N	.	.	.
AWW20564.1	Santiago	I	.	.	.	.	.	.	.	.	.	.	.	.	.	.	G	L	.	N	.	.	.
QBA97753.1	Calvados	I	.	.	.	.	.	.	.	.	.	.	.	.	T	.	R	L	.	N	.	.	.
QBA97752.1	Ardennes	I	.	.	.	.	.	.	.	.	.	.	.	.	T	.	R	L	.	N	.	.	.
QBA97751.1	Pas-de-calais	I	.	.	.	.	.	.	.	.	.	.	.	.	T	.	R	L	.	N	.	.	.
QBA97750.1	Paris	I	.	.	.	.	.	.	.	.	.	.	.	.	T	.	R	L	.	N	.	.	.
WAA38452.1	Brescia	I	.	.	.	.	.	A	T	P	.	.	.	.	.	.	R	L	.	N	.	.	.
ABY81492.1	Miami	.	V	I	M	R	I	A	T	P	V	K	S	Q	.	K	.	.	N	.	E	D	R

**Table 7 animals-14-00598-t007:** NA differences in amino acid terms between the South Africa strain and a group of protein sequences composed of the Miami strain and most recent circulating FC1 strains.

Amino Acid Position		66	67	68	70	72	74	82	123	145	158	159	172	196	199	203	211	248	250	256	258	259	266	269	287	299	304	336	340	341	354	388
USI49342.1	South Africa	T	S	T	K	I	R	N	S	V	K	I	S	N	I	N	I	K	R	D	R	V	S	G	I	I	S	S	N	K	T	R
ABA39858.1	Ohio	.	.	.	.	.	.	.	.	.	.	.	.	.	.	.	.	.	.	.	.	.	.	E	.	.	.	.	.	.	.	.
AWW20361.1	Santiago	I	.	.	.	.	K	.	.	I	.	.	.	.	.	.	.	.	K	N	K	.	.	.	.	.	.	N	.	.	.	.
WCL41104.1	United Kingdom	I	.	.	.	M	K	.	.	I	.	.	.	.	.	.	.	.	K	N	K	.	.	.	.	.	.	N	.	.	.	.
QBA98125.1	Calvados	I	.	.	.	M	K	.	.	I	.	.	.	.	.	.	.	.	K	N	K	.	.	.	.	.	.	N	.	.	.	.
QBA98124.1	Pas-de-calais	I	.	.	.	M	K	.	.	I	.	.	.	.	.	.	.	.	K	N	K	.	.	.	.	.	.	N	.	.	.	.
WAA38453.1	Brescia	I	.	.	.	M	K	.	.	I	.	V	.	.	.	.	.	.	K	N	K	.	.	.	.	.	.	N	.	.	.	.
ABY81495.1	Miami	.	N	I	E	.	E	S	L	.	E	V	A	A	V	H	V	Q	.	.	.	I	N	.	V	V	P	.	S	Q	N	K

## Data Availability

Data are contained within the article.

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
