# Peer review of "First Reported Circulation of Equine Influenza H3N8 Florida Clade 1 Virus in Horses in Italy"

_animals, 2024, doi:10.3390/ani14040598_

Round 1

Reviewer 1 Report

Comments and Suggestions for Authors

The article provides proper data and should be published.

However, the text is too long for the provided amount of information. It should be shortened and published as a communication rather than an article.

For example, data provided in Tables 1 and 2 should be mentioned in the text. Table No. 4 and Figure 1 provide the same data.

Tables 1,2, 3, 5, 6, 7, 8, and 9 may be partially mentioned in the text or published as supplementary material.

The discussion and conclusion are too long and provide well-known general information.

There is no vaccine protecting against infection. The protection level "protected against infection" in Table 4 should be changed.

Author Response

  • The article provides proper data and should be published.

However, the text is too long for the provided amount of information. It should be shortened and published as a communication rather than an article.

Thank you for the comment, but we would like to maintain the structure as a “paper”.

  • For example, data provided in Tables 1 and 2 should be mentioned in the text.

Thank you for the comment, but we think that it is easier to have data in tables, rather than in the text for a better understanding . We think that the data are important, from an epidemiological point of view.

  • Table No. 4 and Figure 1 provide the same data.

Thank you for the advice. It is true, and we decided to maintain the table 4.

  • Tables 1,2, 3, 5, 6, 7, 8, and 9 may be partially mentioned in the text or published as supplementary material.

Thank you for the comment, but we think that it is more clear and easier to understand if data are in tables and in the paper

  • The discussion and conclusion are too long and provide well-known general information

Thank you for the suggestion, we summarized.

  • There is no vaccine protecting against infection. The protection level "protected against infection" in Table 4 should be changed.

The definition "protected against infection" is related to a classification founded in the paper by Cullinane et al, 2020 (doi:10.3390/vaccines8010107).

Reviewer 2 Report

Comments and Suggestions for Authors

There is a need to discuss previous reports of EIV in Italy, including their genetic characterisation. No mention is made of the papers by Martella et al. 2007, Daminiani et al. 2008, or Autorino et al. 2015. For the latter, the paper reports the first case of H3N8 Florida clade 2 in Italy in 2014. It would be meaningful if these papers were also referred to in Introduction as well as Discussion.

A reduction in the content is also warranted.

Line  95 : .. for the detection of the EIV matrix protein

The reference referred to (Spackman et al. 2002) describes an assay for  detection of influenza A viruses, not specifically EIV  (for this aper it was with reference to avian influenza viruses).  

Thus refer here and elsewhere to influenza A virus (IAV) PCR rather than EIV PCR.

Table 2

What does Attitude refer to ? Is it necessary or meaningful to include Saddle in the column ?  Alternatively, is the table as a whole necessary ? Can it rather be briefly summarised in the text ?

Table 4

Include a reference in the legend for the classification of protection levels

Figure 1

Data in Table 4 is adequate, this this figure is not really necessary

Table 8, 9

.. considering as query South African strain

Should this not mean that the South African strain was used as reference ?

For conserved amino acid  relative to the reference sequence, dots may be used  instead

“Amino acid position with a star refers to antigenic human flu virus site”

This sentence doesn’t seem correct. Also, a single amino acid can’t be classed as an antigenic site. Perhaps also give a relevant reference to be able to clarify further.

Line 291.. used as query

Should this not be  “… used as reference” ?

Comments on the Quality of English Language

An improvement in language is needed, to clarify various statements throughout the document.

Line 26-27:  The aim of this study was to…...of imperative value in the

Unclear what is meant here . Need to rephrase and clarify further

Line48: ..that is exceptionally fatal.

Presumably this is meant to mean infrequently or rarely fatal. Adjust accordingly

Line 48- 49:

…could also lead to a lethal exitus … especially in scarce hygienic conditions

Rather  say: …could lead to a lethal outcome ………especially under poor (or suboptimal) hygienic conditions

Line 52

…subtypes of Influenza virus A (family Orthomyxoviridae)

Change to : .. subtypes of influenza A virus  (family Orthomyxoviridae)

Line 73-74

… indicating that EIV circulates

This sentence is unclear.  What is meant by this ? Should it be …. where EIV circulation is suspected ?

Line 78-79

..an economic important sector especially due to public funds, which were dedicated even in 2023

Not clear what is meant. Should it be : ..an economic important sector especially due to significant ongoing public funding .

Line 97….ct<40

…Ct <40

Line 41: haemoagglutinating

 Haemagglutinating

Line 128:  Rstudio

RStudio

Line 165: Molise Region

Molise region

Line 168: classified as riding centre or saddle horses stabled in the owner facility

classed as riding centres, or saddle horses that were stabled at owner facilities

Line 169: The age, attitude and breed…

Not sure attitude is the correct term

Line 170: … reported only one recent (four days be-fore the onset of the clinical signs) introduction of four horses (three reported with clinical signs after the transport) from Cordoba, Spain in the North Italy outbreaks.…

..in the case of outbreaks in North Italy, only one recent  introduction of  four horses from Cordoba was reported, four days prior to the onset of clinical signs in three of the horses.

Line 172: out-breaks

Outbreaks

Line 173: … a traditional horse comingling…….

…a traditional gathering of horses

Line 175: …of horses joined from farms of the whole region suggesting the potential source if the infection

Hundreds of horses from the entire region  gathered at this traditional event , which served as the most likely source of the outbreak

Line 233:  figure 2 and 3

Figure 2 and 3

Line 256: (fig 1)

(Fig. 1)

Tables 6 & 7 : values of the arithmetic and harmonic means

…Marginal likelihood values of the arithmetic and  

Line 330 : a prevalent reaction against specifically for one of the antigens employed

a prevalent reaction against a specific antigen

Line 422…. That are useful to raise the attention level

That are useful in raising awareness of the potential of introducing virus

Line 425: with the equine industry stakeholders

Stakeholders in the equine industry

Line 426: including EI considering frequent

Including EI, considering the frequent….

Author Response

  • There is a need to discuss previous reports of EIV in Italy, including their genetic characterisation. No mention is made of the papers by Martella et al. 2007, Daminiani et al. 2008, or Autorino et al. 2015. For the latter, the paper reports the first case of H3N8 Florida clade 2 in Italy in 2014. It would be meaningful if these papers were also referred to in Introduction as well as Discussion.

We didn’t insert the Italian paper before Autorino et al, 2015 because this last is the more recent paper that discuss about the Italian isolated before.

  • A reduction in the content is also warranted. 

Thank you for the suggestion, we summarized.

  • Line  95 : .. for the detection of the EIV matrix protein

The reference referred to (Spackman et al. 2002) describes an assay for  detection of influenza A viruses, not specifically EIV  (for this aper it was with reference to avian influenza viruses).  

Thus refer here and elsewhere to influenza A virus (IAV) PCR rather than EIV PCR.

Thank you for the suggestion, we changed the target.

  • Table 2

What does Attitude refer to ? Is it necessary or meaningful to include Saddle in the column ?  Alternatively, is the table as a whole necessary ? Can it rather be briefly summarised in the text ?

Thank you for the suggestion, the attitude term refers to the holding category, that was the same for all horses and the column was removed from the table 2, we reported it in the text (line 170).

  • Table 4

Include a reference in the legend for the classification of protection levels

We included

  • Figure 1

Data in Table 4 is adequate, this this figure is not really necessary

We removed it 

  • Table 8, 9

.. considering as query South African strain

Should this not mean that the South African strain was used as reference ?

Should this not mean that the South African strain was used as reference ? ? In Materials and Methods: “The amino acid substitutions were evaluated performing blastp, using as query the South Africa strain (USI49340.1 for HA and USI49342.1 for NA genes) and as subject a group of protein sequences composed of the Miami strain (ABY81492.1 for HA and ABY81495.1 for NA) and most recent circulating FC1 strains, including the one detected in this study”. So, both HA and NA proteins of the South African strain were aligned with blastp against other proteins sequences considered. Indeed the amino acid positions are referred to those of South Africa and that’s why I wrote “considering as query”. I’ve modified with:

Table 8: HA differences in amino acid terms between the South Africa strain and a group of protein sequences composed of the Miami strain and most recent circulating FC1 strains. Amino acid position with a star refers to antigenic human flu virus site (https://doi.org/10.1016/j.virol.2017.02.003 , https://www.ncbi.nlm.nih.gov/pmc/articles/PMC255321/pdf/jvirol00139-0060.pdf)

Table 9: NA differences in amino acid terms between the South Africa strain and a group of protein sequences composed of the Miami strain and most recent circulating FC1 strains

For conserved amino acid  relative to the reference sequence, dots may be used  instead

Thank you, done

“Amino acid position with a star refers to antigenic human flu virus site”

This sentence doesn’t seem correct. Also, a single amino acid can’t be classed as an antigenic site. Perhaps also give a relevant reference to be able to clarify further.

In this article “Genetic evolution of equine influenza virus strains (H3N8) isolated in France from 1967 to 2015 and the implications of several potential pathogenic factors” , in the caption of Table 1 there is written “Amino acid alignments of five antigenic sites A to E between HA sequences determined for French strains (30 strains) and vaccine strains (*) or viral strains causing large-scale outbreaks worldwide (°) compared with A/equine/Miami/1/1963. The antigenic sites defined for human H3 influenza virus was used as a reference (Both et al., 1983).” These references will be included in the caption of Table 8Line 291.. used as query

Should this not be  “… used as reference” ?

We modified query with reference, thank you.

Comments on the Quality of English Language

  • An improvement in language is needed, to clarify various statements throughout the document.
  • Line 26-27:  The aim of this study was to…...of imperative value in the

Unclear what is meant here . Need to rephrase and clarify further

Thank you for the suggestion, we clarified the phrase.

  • Line48:..that is exceptionally fatal.

Presumably this is meant to mean infrequently or rarely fatal. Adjust accordingly

 Thank you for the suggestion, we substituted with rarely.

  • Line 48- 49:

…could also lead to a lethal exitus … especially in scarce hygienic conditions

Rather  say: …could lead to a lethal outcome ………especially under poor (or suboptimal) hygienic conditions

  Thank you for the suggestion, we substituted it

  • Line 52

…subtypes of Influenza virus A (family Orthomyxoviridae)

Change to : .. subtypes of influenza A virus  (family Orthomyxoviridae)

  Thank you for the suggestion, we substituted it

  • Line 73-74

… indicating that EIV circulates

This sentence is unclear.  What is meant by this ? Should it be …. where EIV circulation is suspected ?

 Thank you for the suggestion, we substituted it

  • Line 78-79

..an economic important sector especially due to public funds, which were dedicated even in 2023

Not clear what is meant. Should it be : ..an economic important sector especially due to significant ongoing public funding .

 Thank you for the suggestion, we substituted it

  • Line 97….ct<40

…Ct <40

Thank you for the suggestion, we substituted it

  • Line 41: haemoagglutinating

 Haemagglutinating

Thank you for the suggestion, we substituted it 

  • Line 128:  Rstudio

RStudio

Thank you for the suggestion, we substituted it 

  • Line 165: Molise Region

Molise regionThank you for the suggestion, we substituted it

  • Line 168: classified as riding centre or saddle horses stabled in the owner facility

classed as riding centres, or saddle horses that were stabled at owner facilities

 Thank you for the suggestion, we substituted it

  • Line 169: The age, attitude and breed…

Not sure attitude is the correct term

Thank you for the suggestion, we removed the term, the attitude is the holding category, as specified above. 

  • Line 170: … reported only one recent (four days be-fore the onset of the clinical signs) introduction of four horses (three reported with clinical signs after the transport) from Cordoba, Spain in the North Italy outbreaks.…

..in the case of outbreaks in North Italy, only one recent  introduction of  four horses from Cordoba was reported, four days prior to the onset of clinical signs in three of the horses.

 Thank you for the suggestion, we modified it

  • Line 172: out-breaks

Outbreaks

 Thank you for the suggestion, we modified it

  • Line 173: … a traditional horse comingling…….

…a traditional gathering of horses

 Thank you for the suggestion, we modified it

  • Line 175: …of horses joined from farms of the whole region suggesting the potential source if the infection

Hundreds of horses from the entire region  gathered at this traditional event , which served as the most likely source of the outbreak

Thank you for the suggestion, we modified it 

  • Line 233:  figure 2 and 3

Figure 2 and 3

Thank you for the suggestion, we modified it  

  • Line 256: (fig 1)

(Fig. 1)

Thank you for the suggestion, we modified it

  • Tables 6 & 7 : values of the arithmetic and harmonic means

…Marginal likelihood values of the arithmetic and  …

Thank you for the suggestion, we added it 

  • Line 330 : a prevalent reaction against specifically for one of the antigens employed

a prevalent reaction against a specific antigen

 Thank you for the suggestion, we modified it

  • Line 422…. That are useful to raise the attention level

That are useful in raising awareness of the potential of introducing virus

 Thank you for the suggestion, we modified it

  • Line 425: with the equine industry stakeholders

Stakeholders in the equine industry

 Thank you for the suggestion, we modified it

  • Line 426: including EI considering frequent

Including EI, considering the frequent….

 Thank you for the suggestion, we modified it

Reviewer 3 Report

Comments and Suggestions for Authors

In this work, the authors report for the first time the circulation of the equine influenza H3N8 Florida clade 1 virus in horses in Italy. The importance of increased surveillance and the investigation of vaccination breakdowns in different countries is always emphasized. The tools used in the study for virus characterization and serological studies are appropriate and correctly presented.  

This reviewer highlights, however, the importance of a rapid antigenic characterization. The results should be available soon to allow comparison with strains included in the vaccines. The authors should persuade this reviewer, in abstract and discussion section, the importance of presenting in 2024 the circulation of an equine influenza strain that caused an outbreak in 2019.

Author Response

  • In this work, the authors report for the first time the circulation of the equine influenza H3N8 Florida clade 1 virus in horses in Italy. The importance of increased surveillance and the investigation of vaccination breakdowns in different countries is always emphasized. The tools used in the study for virus characterization and serological studies are appropriate and correctly presented.  
  • This reviewer highlights, however, the importance of a rapid antigenic characterization. The results should be available soon to allow comparison with strains included in the vaccines. The authors should persuade this reviewer, in abstract and discussion section, the importance of presenting in 2024 the circulation of an equine influenza strain that caused an outbreak in 2019.

Thank you for the comment. We apologize for the inconvenience but we took actively part in the Sars-CoV-2 pandemic and some activities, such as collection of data and they publication, were necessarily postponed.

Reviewer 4 Report

Comments and Suggestions for Authors

The primary criticism is that this does not appear to be the first reported circulation of equine influenza H3N8 (Florida virus) in Italy.  The paper that should be referenced which describes more extensive surveillance is below:

Some review of isolates previously genetically characterized in Italy should be mentioned as in: 

Damiani, A. M., Scicluna, M. T., Ciabatti, I., Cardeti, G., Sala, M., Vulcano,

G., et al. (2008). Genetic characterization of equine influenza viruses

isolated in Italy between 1999 and 2005. Virus Res. 131, 100–105.

doi: 10.1016/j.virusres.2007.08.001

Which indicates that H3N8 (Florida lineage) has been previously isolated from horses in Italy

Some additional questions in regard to this paper that would be useful to answer are below...

Line 79…public funds were provided in 2023 for what purpose in the horse industry (would this be used for surveillance, testing?)

What would be the most likely vaccine components for EI used in Italy currently (would they contain only H7N7) for during this time period of the study? 

Is H3N8 circulating in Cardoba Spain and which farm # was exposed to these horses that came from there?

Was the lowest PCR CT for the sample that had virus isolated from it and what were the other Ct values?

Explanation of why other samples could not be sequenced directly from the nasal swab?  Was sequencing only attempted on the 1 sample that was VI positive?

Author Response

  • The primary criticism is that this does not appear to be the first reported circulation of equine influenza H3N8 (Florida virus) in Italy.  The paper that should be referenced which describes more extensive surveillance is below:
  • Some review of isolates previously genetically characterized in Italy should be mentioned as in: 

Damiani, A. M., Scicluna, M. T., Ciabatti, I., Cardeti, G., Sala, M., Vulcano, G., et al. (2008). Genetic characterization of equine influenza viruses isolated in Italy between 1999 and 2005. Virus Res. 131, 100–105. doi: 10.1016/j.virusres.2007.08.001

 Which indicates that H3N8 (Florida lineage) has been previously isolated from horses in Italy

Thank you for the comment. H3N8 Florida lineage has previously been isolated in Italy, but we referred about the first isolation of Florida Clade 1. In fact, as indicated in paper that you mentioned, strains isolated in Italy from 2003 and 2005 were closely related to Newmarket/5/2003, A/eq/Essex/2/2005 and A/eq/Essex/3/2005, initially classified in the American lineage and subsequently classified as belonging to H3N8 Florida Clade 2 as indicated in more recent paper as:

1) “Genetic evolution of equine influenza strains isolated in France from 2005 to 2010” L. J. LEGRAND et al.

Equine Veterinary Journal 47 (2015) 207–211 ISSN 0425-1644 DOI: 10.1111/evj.12244 Equine Veterinary Journal

2) Paillot, R., Pitel, P.H., Pronost, S., Legrand, L., Fougerolle, S., Jourdan, M. and Marcillaud-Pitel, C. (2019) Florida clade 1 equine influenza virus in France. Vet. Rec. 184, 101 (Supplementary Material)

  • Some additional questions in regard to this paper that would be useful to answer are below...
  • Line 79…public funds were provided in 2023 for what purpose in the horse industry (would this be used for surveillance, testing?)

Public fund provided in 2023 have to be allocated for horse racing, not for surveillance (https://www.politicheagricole.it/flex/cm/pages/ServeBLOB.php/L/IT/IDPagina/19844)

  • What would be the most likely vaccine components for EI used in Italy currently (would they contain only H7N7) for during this time period of the study? 

Available vaccines in Italy contains representative strains not only for lineages H7N7 but also H3N8 European and American lineages.

  • Is H3N8 circulating in Cardoba Spain and which farm # was exposed to these horses that came from there?

There is evidence of circulation in Andalusia of H3N8 [https://doi.org/10.1016/j.prevetmed.2018.01.003] but in this specific case the competent authorities have not found relevant information on any further transnational epidemiological investigations.

  • Was the lowest PCR CT for the sample that had virus isolated from it and what were the other Ct values?

Thank you for the question, the Ct values were between 26 (the lowest, from which we obtained the sequences) and 34 (the highest).

  • Explanation of why other samples could not be sequenced directly from the nasal swab?  Was sequencing only attempted on the 1 sample that was VI positive?

Thank you for the comment. Samples are not sequenced directly from the nasal swabs for the low amount of viral copy number recovered, while allantoid fluid collected from inoculated embryonated eggs allowed us to get more material to work with. For genetic analyses were used two samples obtained from the harvesting of the allantoic fluid from the eggs inoculated with the nasal swabs of symptomatic subjects and positive in RRT-PCR but subsequent sequencing was successful for only one sample of these.

Round 2

Reviewer 2 Report

Comments and Suggestions for Authors

1.     In the title, superscript 3 for Giacomo Vincifori does not link to any address details

2.    Author comment: : We didn’t insert the Italian paper before Autorino et al, 2015 because this last is the more recent paper that discuss about the Italian isolated before.

I still believe there is a need to give a brief background on the occurrence of prior EI outbreaks in Italy and what the viral lineages were. The reference to the Autorino 2015 paper is made but no mention as to what the paper indicates as regards EIV in Italy.   A question a reader may ask is what has the EI situation been in Italy prior to the outbreak reported in this paper.

3.     Table 8, Line 296: Amino acid position with a star refers to antigenic human flu virus site

As mentioned previously, a single amino acid can’t be an antigenic site .  Usually 5 or more amino acids make up a linear epitope (this is a better term to use  than antigenic site). The papers referred  to, also don’t indicate this either .  All the antigenic sites (linear epitopes) indicated in the Fougerolle paper, consist of 5 or more amino acids. 

Presumably what is being suggested, is that the amino acid substitution exists within a linear epitope, however, I haven’t been able to find  amino acid 289  within the 5 sites listed in the Fougerolle paper (Table 1).

4.      Summary: Line 19 and elsewhere

 ...were positive by EI RRT-PCR

As mentioned before, it would be better to refer to the assay used as an influenza A virus (IAV) RRT-PCR

Comments on the Quality of English Language

Line 22:  EI circulating strains in relation

EIV circulating strain…

Line 27:  was to genetically characterize EI strains circulating in Italy was to genetically characterize EIV strains circulating in Italy

Line 34:   EI RRT-PCR

IAV RRT-PCR

Line 38:  monitoring circulating EI strains

monitoring circulating EIV strains

Line 52: influenza virus A

Influenza A virus

Line 65: Continuous mutations of influenza viruses,

Ongoing mutations of influenza A viruses

Line 94: The eluate was submitted to reverse-transcriptase real-time (RRT) PCR

The eluate was used for reverse-transcriptase real-time (RRT) PCR

Line 100: presence of a Haemagglutinating virus

..presence of a haemagglutinating virus

Line 157: Fisher's Exact test

Fisher's exact test

Line 166, 213 & 216: Lombardia

Lombardy

Line 166: All the farms belong to the equestrian sector and were classified as riding centres, or saddle horses that were stabled at owner’s facilities.

All the farms belong to the equestrian sector and were classified as riding centres,  or had saddle horses that were stabled at the owners’ facilities.

Line 238:.. filtered by Accession Number (A.N.), Query Coverage (Query cov.)

,.. filtered by accession number (A.N.), query coverage (Query cov.),

Line 268: (fig.2)

(Fig. 2)

Line 296: 19 Amino acid position with a star refers to antigenic human flu virus site [26,27].

Asterisk indicates a.a  located within a linear epitope identified in a human flu virus [26,27].

(Refer to earlier comment. Can’t find this amino acid location in the linear epitope sites described in the publication)

Line 311: usually indicative of the use of vaccines that still contain also this subtype

usually indicative of the use of vaccines that still contain this subtype

Line 315: CT values

Ct values

Line 318: in the only still seronegative horses.

in the remaining seronegative horses.

Line 320: for only one, Brescia/19038317_3/2019, 320 of the two samples for which viral isolation was still successful

for only one of the two samples for which viral isolation was still successful, viz, Brescia/19038317_3/2019,  

Line 327: for both trees, respectively referable to the HA and NA genes.

for both  trees, referable to the HA and NA genes.

Line 328: split into

separate into

Line 334: Table 6 e 7 indicate

Table 6 and 7 indicate

Line 335: keeps a more similar amino acid pro-335 file with that of the Miami strain, in three positions,

..retains a more similar amino acid pro file with that of the Miami strain in three positions,  

Line 337: ….proximity (V285A) of the antigenic human influenza site C

proximity (V285A) to the site C linear epitope of human influenza virus

Line 348: The surveillance for EI is essential both to manage outbreaks and limit virus diffusion, and to: obtain data on the antigenic drift and shift of the circulating strains for vaccine 349 update.

The surveillance for EI is essential to both manage outbreaks and limit viral spread, and to obtain data on the antigenic drift and shift of the circulating strains for updating vaccines.

Line  359: For this, there was a reluctance to submit samples on suspect of this infection, with a consequent underestimation of its circulation; cases reported in our study represent an occasional event in the context of a probably wider viral circulation.

As a result, there has been a reluctance to submit samples in cases where suspicion of infection occurs, leading to a likely underestimation of disease occurrence; thus the cases reported in our study likely represent only a few cases amongst many,  with a more widespread level of viral circulation likely present.

Line 362: is not contemplated in

..is not considered in

 Line 363: laid down however, there are horse competition events that mandatorily require that the participating horses are vaccinated

.. laid down, however, there are equestrian events that require the mandatory vaccination of equine participants

 Line 371: …vaccinated horses showed none to mild clinical signs

.. vaccinated horses showed mild to no clinical signs

Line 372: presence of a cross immunological protection especially in those that had been receiving for several years, a regular vaccination regime.

.. presence of cross protective immunity especially amongst those horses that had been regularly vaccinated over several years

 Line 377: and virus shedding period

… and the duration of viral shedding

 Line 380: in the equid population as also, as already referred to acquire relevant virological and epidemiological informations        1`

….in the equine population, so as to obtain reliable virological and epidemiological information

Line 381: Following humans, the horse is the domestic animal 381 with most frequent movements that underlines

.. after humans, the horse is subjected to the most frequent movement amongst domestic animal

Line 383: to limit their diffusion

.. to limit their spread

Line 390: that are useful in raising awareness of the potential of introducing virus of possible local introduction, together with strains characterization
